# Validation of an Academic Self-Attribution Questionnaire for Primary and Secondary School Students: Implications of Gender and Grade

**DOI:** 10.3390/ijerph19106045

**Published:** 2022-05-16

**Authors:** Ana I. Obregón-Cuesta, Paula Rodríguez-Fernández, Benito León-del-Barco, Santiago Mendo-Lázaro, Luis A. Mínguez-Mínguez, Josefa González-Santos, Jerónimo J. González-Bernal

**Affiliations:** 1Department of Mathematics and Computing, University of Burgos, 09001 Burgos, Spain; aiobregon@ubu.es; 2Faculty of Humanities and Social Sciences, Universidad Isabel I, 09003 Burgos, Spain; 3Department of Psychology and Anthropology, University of Extremadura, 10071 Cáceres, Spain; bleon@unex.es (B.L.-d.-B.); smendo@unex.es (S.M.-L.); 4Department of Education Sciences, University of Burgos, 09001 Burgos, Spain; laminguez@ubu.es; 5Department of Health Sciences, University of Burgos, 09001 Burgos, Spain; mjgonzalez@ubu.es (J.G.-S.); jejavier@ubu.es (J.J.G.-B.)

**Keywords:** attributional styles, educational context, students, primary school, secondary school, gender

## Abstract

The way in which students attribute causes to their successes and failures in school has important implications for their development. The objectives of our research were to validate the Academic Success and Failure Attribution Questionnaire (ASFAQ) and to analyze the gender and grade differences in the ASFAQ data for primary and secondary school students in Spain. For the construction and analysis of the psychometric characteristics of the scale, an exploratory factor analysis (EFA) and a confirmatory factor analysis (CFA) were performed. To compare the ASFAQ scores based on gender and school year, a parametric *t*-test for independent samples and a one-way analysis of variance (ANOVA) was used. A total of 562 students in the fifth (*n* = 228) and sixth year (*n* = 186) of primary studies and the first (*n* = 134) and second year (*n* = 94) of secondary studies participated in the research. The results showed the adequate factorial structure, internal consistency, and validity of the ASFAQ, in addition to statistically significant differences by gender and school year. This research provides scientific evidence about the psychometric properties of the ASFAQ to assess and understand attributional style in the educational context, as well as current and consistent empirical evidence related to gender and grade differences in the attributional patterns of academic success and failure for primary and secondary school students.

## 1. Introduction

Attribution can be defined as the process of assigning causes to one’s own behavior or that of other people. This attribution can be internal, called personal or dispositional attribution, or external, which is known as situational attribution. The development of attributional styles theory is the result of scientific research on the search and assignment of human behavior causes [1].

Weiner’s Attribution Theory [2] is one of the most used in the study of attributional styles. The theory defines academic attribution as the explanation that individuals give for their academic performance in school, whether successful or not [3]. When people attribute the causes of their behavior to factors that do not depend on themselves, they act using an external or situational style of attribution. In contrast, the internal, personal, or dispositional, style is characterized by a causal explanation based on the relationship between the action itself and the result [4,5]. Weiner et al. [2] found in their research that students used the skill, effort, luck and difficulty of the task as the main reasons to explain their academic successes and failures [6].

The attributional style chosen can favor or hinder learning and determines the motivation with which students engage and cope with academic demands, even influencing their perception of themselves and their performance [6]. Its practical implications are increasingly evident [7,8], and school failure can be largely explained by the use of ineffective and maladaptive attributional styles [9]. Since having been identified as important determinants for student learning and self-esteem [10,11,12,13], attributions have been associated with variables such as anxiety and depression or academic performance [14,15,16] and self-efficacy [17].

The academic attributional style acquires special relevance in primary and secondary school students due to their current stage of development [18], and the way in which they attribute causes to their successes and failures in school has important implications for their development [18]. When students attribute their results only to external causes, they make little effort toward their learning and evolution, while if they explain their results with the belief they are lacking in intellectual capacity, their confidence and school performance will be seriously affected [9]. Along the same lines, if you associate success with your own effort or ability, you will probably feel pride and the motivation to continue obtaining good results [17]. Normandeau y Gobeil [19] stated in their study that changes in causal attributions are the result of metacognitive developments in children about themselves as problem solvers and that these determine their emotional reaction and orientation towards the task.

Gender has been shown to play an important role in attributional patterns in the school context. Most of the studies in which gender differences in student attributions have been analyzed report greater self-concept and self-esteem in boys, who frequently attribute their failures to external and unstable causes and their successes to internal causes, as opposed to girls [20,21,22,23]. However, a consistent attributional trend based on gender cannot be affirmed, since it can vary depending on other variables such as the academic content area or the school year [23].

In order to promote adaptive attributional styles, valid and reliable assessment instruments aimed at specific population groups are required. Taking into account the relevance of this aspect to the school context, the main we objective proposed was to validate the Academic Success and Failure Attribution Questionnaire (ASFAQ) with a sample of primary and secondary school students in Spain (Appendix A), hypothesizing that the ASFAQ would be a rigorous scientific instrument with the adequate validity and reliability to evaluate the attributions of academic success and failure in primary and secondary school students. In addition, we intended to study gender and school year differences in order to provide current and consistent empirical evidence. Along these lines, and considering the findings of previous studies, we expected to find attributional differences based on gender and school year.

## 2. Materials and Methods

### 2.1. Participants

The sample consisted of 562 students of Compulsory Primary Education (CPE) and Compulsory Secondary Education (CSE), aged between 10 and 14 years (mean (M) = 11.66; standard deviation (SD) = 1.21). A total of 284 subjects belonged to the male gender and 278 to the female. The CPE students (*n* = 334) were in their fifth (*n* = 228) and sixth school years (*n* = 186), and those in CSE (*n* = 148) belonged to the first (*n* = 134) and second school years (*n* = 94). The sample was collected from 5 different schools, both public (*n* = 4) and private (*n* = 1), in the Autonomous Community of Castilla y León.

The selection of the sample was performed by conglomerates.

### 2.2. Instruments

The Academic Success and Failure Attribution Scales were designed with the objective of evaluating the attributions of academic success and failure. The scales consist of twelve items in Likert format with five intervals in numerical form from 1 (Not at all agree) to 5 (Totally agree). The items aim to adequately show the most relevant contents of the construct for evaluation. In this sense, students attribute their successes and failures mainly to four elements: ability, effort, task difficulty and luck; classified according to three dimensions: locus of causality, stability and controllability [24]. The locus of causality can be internal (skill, effort) or external (chance, difficulty of the task) and the causes can be stable (ability) or unstable (effort, luck) [25].

### 2.3. Procedure

Following the ethical guidelines of the American Psychological Association regarding consent, confidentiality and anonymity in the answers, a member of the research group contacted school principals and explained to them the objectives of the research.

Not all the schools contacted decided to collaborate in the research. Those who did not participate indicated a lack of time in the classroom and the difficulty of obtaining parental consent as causes.

Once the collaboration was accepted, the participants were contacted in the classrooms, and after obtaining their parents’ or legal guardians’ informed consent, they proceeded to fill in the scales. Its completion was anonymous, guaranteeing the confidentiality of the data obtained and its exclusive use for research purposes. The administration of the scales was performed during school hours, with explanations as to how they should complete it and the answering of any questions that arose during the process. We insisted the nature of the investigation be kept anonymous. The questionnaires were completed individually in a suitable environment and without distractions. The process of completing the questionnaires lasted approximately 15 min.

No questionnaire was rejected.

All the students in the grades selected by the principals were included in the research. No boy or girl was excluded based on their culture, language, religion, race, disability, sexual orientation, ethnicity, gender or age.

The Bioethics Committee of the University of Burgos approved the research (Reference UBU 032/2021), respecting all the requirements established in the Declaration of Helsinki of 1975.

### 2.4. Data Analysis

Initially, for the construction and analysis of the psychometric characteristics of the scales, an exploratory factor analysis (EFA) was performed. Once the EFA was performed, the factorial structure found was confirmed by a confirmatory factor analysis (CFA). The reliability of the scale factors was calculated using Cronbach’s alpha, the Composite Reliability coefficients, McDonald’s Omega and the Extracted Mean Variance. The EFA was performed using the SPSS-21 program; for the CFA, the AMOS-21 program was used.

To compare the scores obtained in the ASFAQ on the basis of gender, a parametric *t*-test for independent samples was used, while the analysis on the basis of school year was performed using the one-way ANOVA test. A statistical significance value of *p* < 0.05 was established, using SPSS version 25 software (IBM-Inc., Chicago, IL, USA).

## 3. Results

The original sample (*n* = 562) was divided into two randomly drawn subsamples (n1 = 276 and n2 = 276). The first (n1) was used to perform the EFA and the second (n2) as a validation sample for the CFA. Both subsamples were equivalent based on gender, χ^2^(1) = 0.359, *p* = 0.549, and age, t (560) = 0.285, *p* = 0.776

### 3.1. Exploratory Factor Analysis

Two exploratory analyses were performed for the Attribution Scales of success and failure, respectively.

#### 3.1.1. Exploratory Factor Analysis of the Attributions of Academic Success

The sample adequacy measure (Kaiser–Meyer–Olkin test = 0.828) and Bartlett’s sphericity test (χ^2^ = 1826.282 (66), *p* < 0.001) justified the factor analysis. Using Kaiser’s rule [26] eigenvalues greater than unity and the unweighted least squares extraction method with Varimax rotation, a solution of three factors was obtained (Table 1) that together explained 55.9% of the variance. The first factor, “Controllable internal attributions” (four items), explained 21.8% of the variance and the collected information on the attributions with an internal location, in which the students were able to exert some influence or control over the cause to which the problem was attributed. The second factor, “Uncontrollable internal attributions” (four items), explained 21% of the variance and the collected information on the attributions with an internal location in which people could not exert influence or control over the cause to which the problem was attributed. The third factor, “Uncontrollable external attributions” (four items), explained 13.1% of the variance and the collected information related to the attributions with an external location in which people could not exert influence or control over the cause to which the problem was attributed.

Cronbach’s alpha of factors 1 (α = 0.744), 2 (α = 0.781) and 3 (α = 0.734) demonstrated an acceptable level of internal consistency.

#### 3.1.2. Exploratory Factor Analysis of the Attributions of Academic Failure

The sample adequacy measure (KMO = 0.902) and Bartlett’s test of sphericity (χ^2^ = 3566.131 (66), *p* < 0.001) justified the factor analysis. Using Kaiser’s rule eigenvalues greater than unity and the unweighted least squares extraction method with Varimax rotation, a solution of three factors was obtained (Table 2) that together explained 70.2% of the variance. The first factor, “Controllable internal attributions” (four items), explained 24.5% of the variance and the collected information on the attributions with an internal location in which people could exert some influence or control over the cause to which the problem was attributed. The second factor, “Uncontrollable internal attributions” (four items), explained 23.5% of the variance and the collected information on the attributions with an internal location in which people could not exert influence or control over the cause to which the problem is attributed. The third factor, “Uncontrollable external attributions” (four items), explained 22.2% of the variance and the collected information about attributions with an external location in which people could not exert influence or control over the cause to which it was attributed.

Cronbach’s alpha of factors 1 (α = 0.876), 2 (α = 0.848) and 3 (α = 0.827) indicated an adequate level of internal consistency.

### 3.2. Confirmatory Factor Analysis

The CFA for each attribution scale (success and failure) was performed with the second subsample (n2 = 276) with the aim of confirming the structures of three factors found in the AFE and whether they were related or independent.

Taking into account some of the most-used fit indices (χ^2^, χ^2^/gl, Comparative Fit Index (CFI), Tucker-Lewis Index (TLI), Root Mean Square Error of Approximation (RMSEA) and Standarized Root Mean-Square (SRMR)) using the maximum likelihood method, four models of attributions for success and four of attributions for failure. They are as follows: M1: Three related factors; M2: three first order factors and one second order factor that brings together the first two factors; M3 three independent factors; M4: a single factor. The χ^2^ should acquire non-significant values (*p* > 0.05), the χ^2^/gl was considered acceptable when it was less than 5; values greater than 0.90 of the incremental indices (CFI and TLI) and ≥0.08 of the RMSEA [27] and SRMR [28] were considered acceptable [29].

#### 3.2.1. Confirmatory Factor Analysis of Success Attributions

All the successful attribution models (Table 3) presented a significant chi^2^ value (*p* < 0.05). Models 3 and 4 were discarded, since the significant chi^2^ square values and the CFI, TLI, RMSEA and SRMR indices did not approach the ideal values. The CFI, TLI indices of models 1 and 2 presented values above 0.92; model 1 presented the best fit, with a lower value of χ^2^/gl, higher CFI and TLI fit indices and lower values of the RMSEA and SRMR indicators (Table 3).

The *t*-values (ranging from 8.38 to 16.76) of the non-standardized regression coefficients of model 1 were statistically significant. The ranges of the standardized coefficients for factor one (0.650–0.834), two (0.604–0.803) and three (0.601–0.781) demonstrated the consistency of the indicators for the measurement of the constructs, these being clearly related (Figure 1).

In addition, the coefficients of the Extracted Mean Variance (EMV and Composite Reliability (CR), with values greater than 0.50 of EMV and 0.80 of CR, showed evidence of reliability in the model for three related factors: (F1 [four items]: EMV = 0.519, CR = 0.811; F2 [four items]: EMV = 0.524, CR = 0.813; F3 [four items]: EMV = 0.509, EMV = 0.804.

#### 3.2.2. Confirmatory Factor Analysis of Failure Attributions

All failure attribution models (Table 3) present a significant chi^2^ value (*p* < 0.05). Again, models 3 and 4 were discarded, since the CFI, TLI, RMSEA and SRMR indices did not approach ideal values. The CFI, TLI indices of models 1 and 2 presented values above 0.95; Model 1 presented a better fit with a lower value of χ^2^/gl, higher CFI and TLI fit indices, and lower values of the RMSEA and SRMR indicators (Table 4).

The *t*-values (ranging from 15.31 to 20.40) of the non-standardized regression coefficients of model 1 were statistically significant. The ranges of the standardized coefficients for factor one (0.725–0.836), two (0.694–0.865) and three (0.720–0.870) were related in a statistically significant way, demonstrating the consistency of the indicators for the measure of the constructs (Figure 2).

### 3.3. Attributional Differences Based on Gender and School Year

#### 3.3.1. Analysis of the ASFAQ According to Gender

The results of the inferential analysis showed statistically significant differences in the attributions of academic success according to gender, but not in those of academic failure (Table 5). More specifically, the male participants demonstrated a greater attribution of their academic success to uncontrollable internal causes (*p* <.001) such as intelligence, good memory, talent or calm character; as well as external causes (*p* <.001) such as easy exams, good luck, low demands or good explanations from teachers. No significant differences were found in the controllable internal attributions of academic success according to gender (*p* = 0.057).

#### 3.3.2. Analysis of the ASFAQ According to the School Year

Statistically significant differences were found depending on the academic year in all the factors (Table 6). In general, the attributions of academic success decreased as the academic year increased. The fifth grade primary students reported the highest average scores in the controllable and uncontrollable internal and external attributions, followed by the sixth grade primary students, first grade secondary students, and finally, second grade secondary students.

Regarding the attributions of academic failure, statistically significant differences were found in the internal attributions between the students in fifth grade primary and second grade of secondary school (*p* = 0.001); The latter demonstrating a greater attribution of academic failure to aspects such as spending little time preparing for exams, making little effort in class, not paying attention in class, or not using appropriate strategies to prepare for exams. Statistically significant differences were also obtained in the external attributions of academic failure, where the students in the first cycle of secondary school tended to attribute their failures to aspects such as bad luck or the difficulty of the exams.

## 4. Discussion

The main objective of this study was to validate a questionnaire designed to evaluate the attributions of academic success and failure in primary and secondary school students. The relevance of this study comes from the need to develop an instrument tailored to this specific population that considers the school context, since the way in which students attribute causes to their successes and failures during this stage has been shown to have important implications in their development [18], as well as in academic and social results [14,15,18,19,30].

The results of the exploratory and confirmatory factor analysis performed revealed the adequate factorial structure as well the internal consistency and validity of the instrument. In addition, the ASFAQ is invariant by gender, so it is an appropriate instrument with which to evaluate the attributions of academic success and failure in primary and secondary school students.

To date, no specific instrument has been validated that analyzes the attributional styles of primary and secondary school children and adolescents in a Spanish academic context, though this would entail great benefits in a fundamental stage of their evolutionary development. Most refer to the general population or university students [31]. Rotter [32] designed the Locus of Control Scale (LOC), one of the most widely used instruments created to assess the attributional style in groups of students, composed of 29 items which measure individual differences in generalized expectations for internal and external control. There are other scales aimed at the university academic field, such as the Multidimensional-Multiattributional Causality Scale [33], Attributional Style Questionnaire [34], Multidimensional Attribution Scale [35], Academic Attributional Style Questionnaire (AASQ) [30], Sydney Attribution Scale [36] and the General Achievement Attributional Motivation Scale [37]. Similarly, for adolescent students, there are scales such as an adaptation of the Strategy and Attribution Questionnaire [31].

In this sense, the main contribution of this project is that it enables the valid and reliable evaluation of the attributions of academic success and failure in the educational context of primary and secondary school, providing useful information for the improvement of variables such as motivation, self-efficacy or academic performance, and even the improvement of learning approaches and students’ regulation [1]. Along these lines, if educational failure is explained in terms of ineffective learning or attributional strategies, the student will be able to select or develop more effective ones [9].

Previous research with university students found a statistically significant positive relationship between adaptive attributional styles and variables such as academic performance, orientation to the learning goal, and academic self-concept [38,39]. Furthermore, psycho-emotional variables related to anxiety and depression have been associated with external attributional patterns in positive situations and with internal and stable attributional patterns in negative situations [40]. Haller et al. [41] found in their study that adolescents with anxiety had a significant tendency to interpret situations more negatively. When compared to those without anxiety; students with anxious and depressive symptoms were prone to develop maladaptive attributional styles that triggered cognitive interference in attention and processing in view of the perception of a threat [42]; they also experienced the presence of interpretive biases in threatening [43] or ambiguous social contexts [44] and less executive functioning [45].

Regarding the second objective of the research, based on the analysis of gender and grade differences in the attributions of academic success and failure, the results showed that boys attributed their academic successes to uncontrollable internal causes (intelligence, good memory, talent) or external causes (easy exams, good luck, undemanding class work); but no differences were found with respect to girls in controllable internal attributions such as effort or dedication, or in attributions for academic failure. These results coincide with those obtained by Almeida et al. [20], where the male students attributed their good academic results to their abilities and the female students to their effort and personal work. For their part, Inglés et al. [46] found that the attributional style of the male students is somewhat more adaptive than that of the female students, since they attribute success to ability (internal cause, stable and uncontrollable) and failures to lack of effort (internal cause, unstable and controllable), which tends to preserve their self-esteem and self-concept [20].

Although most empirical research aimed at analyzing gender differences in students’ attributional patterns supports these findings [46], factors such as age and academic year have also been shown to have a significant implication in academic self-attributions [23,47,48,49]. Students in lower courses tend to attribute their academic results to external causes significantly more, such as study methods or the teacher, and they take more responsibility for their achievements and failures as they progress through the course, attributing them more to internal causes [23,48,49,50]. These results partially coincide with those obtained in the present investigation, where students from lower grades reported a greater tendency to attribute their academic successes to external and internal causes; and students from higher courses attributed their academic failures to internal controllable and external causes the most. Inglés et al. [46] showed a mixed attributional pattern in the lower grade students with respect to the rest of the grades, as they attributed their successes to a greater extent to controllable internal and external causes. These findings are in line with those obtained in this study, with lower grade students being the ones who most explained their successes by appealing to both internal and external causal factors.

The statistically significant differences for gender and academic year in students’ attributional patterns reiterate the importance of considering and promoting adaptive attributional patterns in the educational setting, characterized by attributions of successes to internal causes such as ability and effort and of failures to unstable causes such as a lack of effort or bad luck [9]. For this, the ASFAQ facilitates a correct evaluation of the attributions of academic success and failure and a greater controllability and malleability of the attributional strategies and styles in the educational context of primary and secondary school. Although the ASFAQ presents sufficient evidence for validity and reliability, it has limitations, such as the difficulty in generalizing the results to other population groups, which compromises the external validity (population and ecological) of the questionnaire, as well as establishing validity evidence that is convergent and discriminant. That is why the replication and expansion of the study, in addition to confirming the factorial structure, would allow for the establishment of evidence for convergent and discriminant validity.

## 5. Conclusions

This study demonstrates the psychometric properties of the ASFAQ for the evaluation of attributions of academic success and failure. In this way, the present research provides a scientific and rigorous instrument, with adequate validity and reliability, to determine the attributional styles used within the educational context of primary and secondary school and promote adaptive attitudes and behaviors that favor learning. In addition, the statistically significant differences in gender and academic year that were found in this research provide updated and relevant information for the development of psychoeducational interventions aimed at correcting students’ maladaptive attributional styles.

The limitation of the research lies in the fact that it was carried out on a sample of students from a single geographical region, the autonomous community of Castilla and Leon in Spain.

Future researchers should analyze the ASFAQ factor structure within different student populations, with the aim of improving the tool. In addition, future studies should analyze the structure of invariance with respect to other variables such as region, county, center size or type of center (e.g., private, public or concerted). Finally, it would be especially interesting to carry out a longitudinal study with the same students over the course of several academic years.

## Figures and Tables

**Figure 1 ijerph-19-06045-f001:**
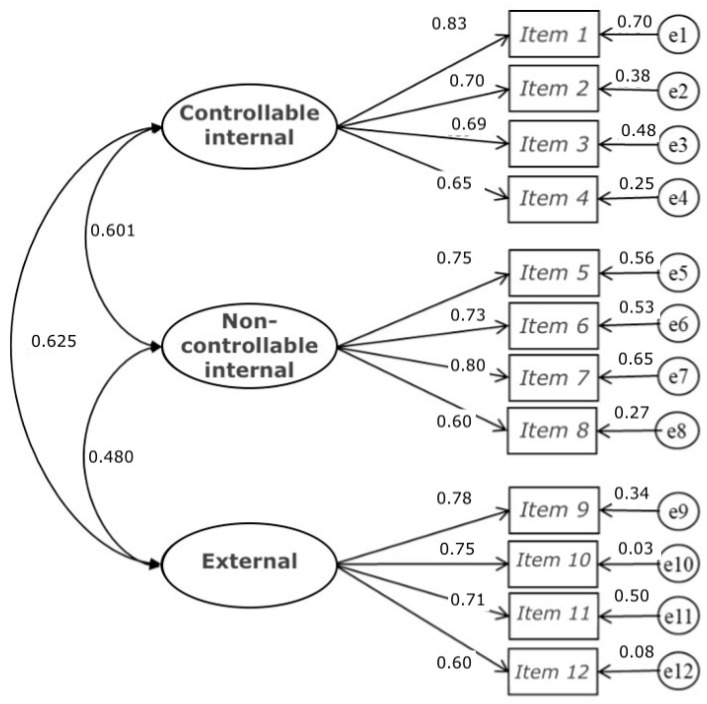
Model of three related factors of the Scale of Attributions of Academic Success.

**Figure 2 ijerph-19-06045-f002:**
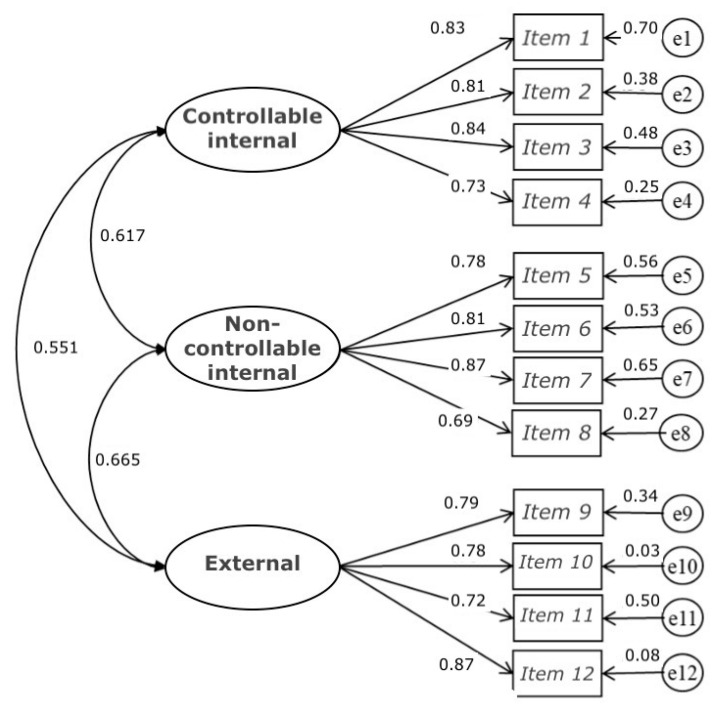
Model of three related factors of the Scale of Attributions of the Academic Failure.

**Table 1 ijerph-19-06045-t001:** Exploratory factor analysis of the “Academic Success Attribution Scale”.

	Component			Communalities
	CIA	UIA	UEA	
I pass because I try so hard in class	**0.798**	0.231	0.056	0.693
I pass because I spend a lot of time preparing for the exams	**0.722**	0.127	−0.009	0.538
I pass because I pay a lot of attention in the classes	**0.603**	0.330	0.018	0.473
I pass because I use some strategy to prepare for the exams (I organize, summarize, review, memorize the topics)	**0.662**	0.166	−0.049	0.468
I pass because I am very intelligent	0.228	**0.780**	0.068	0.665
I pass because I have a very good memory	0.178	**0.795**	0.017	0.664
I pass because I have a lot of talent, that is, I have a lot of natural capacity for studies	0.246	**0.802**	0.063	0.707
I pass because I have a calm character and I don’t get nervous in the exams	0.145	**0.616**	0.169	0.430
I pass because the teachers give easy exams	0.369	0.129	**0.615**	0.531
I pass because I have good luck	−0.145	0.152	**0.723**	0.566
I pass because the level of demand in my class is very low	0.044	−0.011	**0.734**	0.541
I pass because my teachers explain the topics very well	0.291	0.132	**0.609**	0.438

CIA = Controllable internal attributions; UIA = Uncontrollable internal attributions; UEA = Uncontrollable external attributions. Extraction method: Maximum likelihood. Rotation method: Varimax normalization.

**Table 2 ijerph-19-06045-t002:** Exploratory factor analysis of the “Academic Failure Attribution Scale”.

	Component			Communalities
	CIA	UIA	UEA	
I fail because I make little effort in class	**0.823**	0.189	0.199	0.752
because I spend little time preparing for exams	**0.843**	0.214	0.124	0.771
I fail because I pay little attention in class	**0.785**	0.233	0.273	0.745
I fail because I don’t use strategies to prepare for exams (organize, summarize, review, memorize topics)	**0.775**	0.240	0.115	0.672
I fail because I’m not very smart	0.191	**0.820**	0.183	0.742
I fail because I don’t have a good memory	0.248	**0.774**	0.257	0.727
I fail because I have little talent, that is, I have little natural capacity for studies	0.238	**0.835**	0.215	0.800
I fail because I have a nervous character and I cannot calm down in the exams	0.261	**0.624**	0.283	0.538
I fail because the teachers give difficult tests	0.143	0.159	**0.834**	0.742
I fail because I have bad luck	0.140	0.327	**0.659**	0.562
I fail because my teachers do not explain the subjects well	0.182	0.127	**0.789**	0.672
I fail because the level of demand in my class is very high	0.209	0.331	**0.736**	0.696

CIA = Controllable internal attributions; UIA = Uncontrollable internal attributions; UEA = Uncontrollable external attributions. Extraction method: Maximum likelihood. Rotation method: Varimax normalization.

**Table 3 ijerph-19-06045-t003:** Goodness-of-fit indices of the proposed academic success attribution models.

Models		χ^2^	CMIN/DF	CFI	TLI	RMSEA	SRMR
**M1**	3 related factors	*p* < 0.001	2.490	0.961	0.945	0.052	0.043
**M2**	3 factors of 1st order and 1 of 2nd order	*p* < 0.001	3.072	0.949	0.923	0.082	0.047
**M3**	3 independent factors	*p* < 0.001	7.717	0.811	0.751	0.170	0.182
**M4**	1 unique factor	*p* < 0.001	8.281	0.796	0.730	0.114	0.073

CMIN/DF = ratio of chi^2^ over degrees of freedom; CFI = comparative fit index; TLI = Tucker–Lewis index; RMSEA = root mean square error of approximation; SRMR = standardized residual mean square root.

**Table 4 ijerph-19-06045-t004:** Goodness-of-fit indices of the proposed academic failure attribution models.

Models		χ^2^	CMIN/DF	CFI	TLI	RMSEA	SRMR
**M1**	3 related factors	*p* < 0.001	2.391	0.981	0.974	0.050	0.034
**M2**	3 factors of 1st order and 1 of 2nd order	*p* < 0.001	2.859	0.977	0.965	0.058	0.038
**M3**	3 independent factors	*p* < 0.001	11.345	0.842	0.8071	0.136	0.294
**M4**	1 unique factor	*p* < 0.001	19.044	0.724	0.725	0.179	0.100

CMIN/DF = ratio of chi^2^ over degrees of freedom; CFI = comparative fit index; TLI = Tucker–Lewis index; RMSEA = root mean square error of approximation; SRMR = standardized residual mean square root.

**Table 5 ijerph-19-06045-t005:** Results of the *t*-test for independent samples between the ASFAQ and gender.

ASFAQ		Gender				*T*-Test for Independent Samples	
		Male (*n* = 284)		Female (*n* = 278)			
		M	SD	M	SD	t	*p* Value (Bilateral)
**Attributions of academic success**	**Controllable internals**	14.743	3.210	15.277	3.423	−1.907	0.057
	**Non-controllable internals**	13.472	3.691	12.025	3.973	4.514	<0.001
	**External**	10.901	2.895	10.058	2.900	3.451	<0.001
**Attributions of academic failure**	**Controllable internals**	8.109	4.464	7.413	4.066	1.932	0.054
	**Non-controllable internals**	7.809	4.143	8.262	4.341	−1.264	0.206
	**External**	7.753	3.878	7.305	3.635	1.412	0.159

SD = Standard Deviation; ASFAQ = Academic Success and Failure Attribution Questionnaire.

**Table 6 ijerph-19-06045-t006:** Results of the ANOVA test between ASFAQ and the school year.

ASFAQ		School Year								ANOVA Test	
		5° CPE (*n* = 148)		6° CPE (*n* = 186)		1° CSE (*n* = 134)		2° CSE (*n* = 94)			
		M	SD	M	SD	M	SD	M	SD	F	*p* Value
**Success**	**Controllable internals**	16.148 ^b,c^	2.898	15.623 ^d,e^	3.118	14.313 ^b,d,f^	3.056	12.978 ^c,e,f^	3.606	24.229	<0.001
	**Non-controllable internals**	14.047 ^b,c^	3.666	13.193 ^d,e^	3.595	11.880 ^b,d^	3.893	11.106 ^c,e^	3.808	15.415	<0.001
	**Externals**	11.695 ^a,b,c^	3.002	10.854 ^a,d,e^	2.692	9.619 ^b,d^	2.830	9.074 ^c,e^	2.415	23.058	<0.001
**Failure**	**Controllable internals**	7.317 ^c^	4.505	7.505	4.454	7.873	3.890	8.829 ^c^	3.974	2.758	0.042
	**Non-controllable internals**	7.506	4.425	7.661	4.131	8.664	4.251	8.702	4.020	3.033	0.029
	**Externals**	6.925 ^b,c^	3.945	6.430 ^d,e^	3.043	8.835 ^b,d^	4.084	8.808 ^c,e^	3.383	1.883	<0.001

SD = Standard Deviation; ASFAQ = Academic Success and Failure Attribution Questionnaire; CPE = Compulsory Primary Education; CSE = Compulsory Secondary Education. ^a^ Sig < 0.05 in the post-hoc analysis (Bonferroni) between 5th and 6th CPE; ^b^ between 5th CPE and 1st CSE; ^c^ between 5th CPE and 2nd CSE; ^d^ between 6th CPE and 1st CSE; ^e^ between 6th CPE and 2nd CSE; ^f^ between 1st and 2nd CSE.

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
