# Peer review of "Validation of an Academic Self-Attribution Questionnaire for Primary and Secondary School Students: Implications of Gender and Grade"

_ijerph, 2022, doi:10.3390/ijerph19106045_

Round 1
Reviewer 1 Report
While your research is relevant and interesting, your methods are sound, and your results and conclusions were well-explained, frequent errors on the first two pages of the manuscript (lines 2-93) diminished the quality of the presentation.
First, the word "in" used in the title, Validation of an academic self-attribution questionnaire in primary and secondary school students: Implications of gender and grade, should be "for". The measurement instrument was validated for the purpose of using it for (or with), not in students. (Please refer to https://onlinelibrary.wiley.com/doi/abs/10.1111/j.1746-1561.2007.00170.x , https://www.tandfonline.com/doi/abs/10.1080/09286580701772029 , and https://repositorio.grial.eu/bitstream/grial/527/1/SemesterofCode_Students_Questionnaire.pdf )
Other errors on the first page of the manuscript are as follows.
- Line 18: Change "The objective of this research was to..." to "The objectives of this research were to..." (You have two objectives.)
- Lines 19-20: Insert "ASFAQ data for" between "analyze gender
der and grade differences in" and "primary and secondary school students in Spain." (Obviously, there are many gender and grade level differences for students in every country; be sure readers know that the focus of your study was to determine differences for the variables measured by the ASFAQ.) - Lines 21-22: "Exploratory Factor Analysis" and "Confirmatory Factor Analysis" are not capitalized.
- Lines 23-24: Change "the parametric T test" to "a parametric t-test" and change "the one-way ANOVA test" to "a one-way ANOVA". (Note the correct use of articles - a, an, the - as well as when to omit articles.)
- Line 24: Change "belonging to" to "in". ("In" is more appropriate here, and you should avoid unnecessary wording in abstracts.)
- Line 24: The first sentence in "(3) Results" is not a result; it describes your sample.
- Lines 17-31: I strongly recommend that you completely revise your abstract, omitting numbers and subheadings [i.e., "(1) Background", " (2) Methods", "(3) Results", "(4) Conclusions"]. Readers will understand that you are describing the background, methods, results, and conclusions as they read your abstract. It is not necessary to point this out to them.
- Lines 35-38: The meaning of this first sentence is somewhat lost due to questionable wording.
- I would argue that human beings do not frequently face a constant search and assignment of causes of their own and other people's behavior. Rather, humans search for and assign causes because they are curious about behavior.
- The rise of attributional styles did not result from human beings facing "a constant search and assignment of causes", which is what you seem to be implying. Attributional styles theory resulted from scientific research about human behavior, which is related to searching and assigning. (This opening sentence is extremely important and must be flawless. I suggest a complete rewrite.)
- Lines 39-41: The sentence, "Weiner's Attribution Theory [2] is one of the most used in its study and defines academic attribution as the explanation that an individual gives about the reasons for his academic performance in school, whether successful or not, once occurred" should also be rewritten.
- Replace "its study" with "the study of attributional styles".
- Break this long sentence into two sentences by replacing "and" on line 39 with a period. Begin the next sentence with "The theory defines academic attribution..." (There are other long, meandering sentences in your introduction; they are confusing and nonsensical at times. Break up long sentences into two or more sentences so the reader will better understand your meaning.)
- Delete "about the reasons". (Explanations typically include reasons, so this is redundant.)
- Replace "an individual gives" with "individuals give" and replace "his" with "their". (The use of gender-neutral pronouns is extremely important. Please fix this throughout your manuscript.)
- Delete "once occurred". (It is unnecessary.)
Approximately 60% of the lines from 2 through 93 contain at least one mistake in English usage (e.g., capitalization, punctuation, verb tense, use of articles, misused words), which would be a significant distraction to readers who are interested in this work. Although I recommended "Accept after minor revision", the many mistakes in English usage will require you to make major revisions to the manuscript.
Reviewer 2 Report
Review the attached document

Reviewer 3 Report
This is a well written study: the topic is well described with sufficient literature, the grammar is well structured and the manuscript reads well.
